# Continuity of Operations in Newborn Screening: Lessons Learned from Three Incidents

**DOI:** 10.3390/ijns10030055

**Published:** 2024-08-01

**Authors:** M. Christine Dorley, Elizabeth Bair, Patricia Ryland, Amanda D. Ingram, Emily Reeves, Kara J. Levinson, Ona O. Adair, Jenny F. Meredith, Susanne Crowe

**Affiliations:** 1Division of Laboratory Services, Tennessee Department of Health, Nashville, TN 37216, USA; kara.levinson@tn.gov; 2Public Health Laboratory, South Carolina Department of Public Health, Columbia, SC 29223, USA; bairea@dph.sc.gov (E.B.); adairoo@dph.sc.gov (O.O.A.); meredijf@dph.sc.gov (J.F.M.); 3Florida Bureau of Public Health Laboratories, Jacksonville, FL 32202, USA; patricia.ryland@flhealth.gov (P.R.); susanne.crowe@flhealth.gov (S.C.); 4Division of Family Health and Wellness, Tennessee Department of Health, Nashville, TN 37216, USA; amanda.d.ingram@tn.gov; 5Florida Newborn Screening Follow-Up Program, Tallahassee, FL 32301, USA; emily.reeves@flhealth.gov

**Keywords:** contingency planning, continuity of operations, emergency preparedness, disaster, incident, newborn screening

## Abstract

Three incidents that impacted two US newborn screening (NBS) programs highlight the importance of contingency planning for the continuity of operations (COOP). Other NBS programs may benefit from the experience of these state programs for their own contingency planning efforts. Through after-action reviews conducted post-incident, crucial elements for the successful management of an incident were identified. We detailed the strengths, weaknesses, improvements needed, and future actions that will assist in preparing for other incidents as lessons learned.

## 1. Introduction

Public health laboratories (PHLs) ensure or provide core functions such as newborn screening (NBS) to support and maintain population health [1,2]. NBS consists of screening for hearing and critical congenital heart defect (CCHD) and the laboratory testing of dried blood spot (DBS) specimens to identify infants at risk of metabolic, endocrine, and genetic diseases [3]. Any disruptions in the NBS process can pose a serious threat to an affected newborn [4]. Therefore, it is important that PHLs and, specifically, NBS programs engage in contingency planning to maintain the continuity of operations (COOP) in any circumstance [5,6,7,8,9,10]. While hearing and CCHD are vital components of the NBS system, the focus of this manuscript is on the laboratory and follow-up processes for contingency planning and maintaining the COOP.

Contingency planning can be complicated due to differences in the testing panel from state to state, differences in daily specimen volumes, days and hours of operation, differences in NBS follow-up processes, and the time commitment that is involved. Despite these challenges, Florida (FL), South Carolina (SC), and Tennessee (TN) NBS programs (hereafter collectively represented as “we” or “our”) have developed reciprocal relationships to provide support when one of the programs experiences an incident. In this manuscript, we use “incident” to describe disruptions to NBS operations in TN and SC. An incident is “an occurrence, either human-caused or naturally occurring, that requires action to prevent or minimize loss of life or damage to property or natural resources” [11]. An incident describes “… any scenario, threat, disaster, or other public health emergency” [11]. Based on this definition, an incident is most appropriate in the context of our experiences. We detail three incidents occurring over 2020–2023: one act of domestic terrorism, one internet failure, and one due to a crucial instrument requiring repairs.

### 1.1. Incident #1: Domestic Terrorism

On Christmas day, 2020, an explosion damaged the AT&T building in Nashville, crippling phone and internet services with regional impacts. Because the state of TN is an AT&T customer, mid-state offices, including the TN Department of Health Division of Laboratory Services and the NBS laboratory (TNDLS), experienced a complete internet outage. On 26 December, staff were unable to access the laboratory information management system (LIMS); therefore, they could not assign accession numbers to specimens or perform any testing. Later that evening, the TN Health Alert Network alerted staff that mid-state offices would be indefinitely closed. The laboratory director activated the COOP with the FL Bureau of Public Health Laboratory in Jacksonville (FL BPHL). Subsequently, the FL BPHL tested 1109 DBS specimens for TNDLS. State offices reopened on the fourth day after the bombing, and the TNDLS resumed normal operations.

### 1.2. Incident #2: Internet Failure

On 27 May 2021, the SC Public Health Laboratory (PHL) began to experience a disruption to network connectivity. The disruption was intermittent and worsened the following day until the SC Newborn Screening Laboratory (NBSL) could no longer process DBS specimens for testing. The NBSL activated its COOP and sent specimens to the TNDLS beginning on Saturday, 29 May, through Friday, 4 June, when the COOP was deactivated. TNDLS tested 918 specimens for the SC NBSL. The root cause of the intermittent disruption to network connectivity and the subsequent outage was a failed part equal in size to a small cell phone. This part was in a mainframe computer located several miles from the SC PHL. 

### 1.3. Incident #3: Crucial Instrument Requiring Repairs

On 20 September 2023, the tandem mass spectrometer (MS/MS) used for testing SC NBS specimens for amino acids, acylcarnitines, and succinylacetone was inoperable. A second MS/MS, which served the purpose of redundancy, was also not functional. Due to the age of the two MS/MS instruments, the parts needed for the repairs could not be immediately located. When it became apparent that the instrumentation would not be repaired for several days, the SC NBSL activated its COOP and sent specimens to the TNDLS for seven days, beginning on Wednesday, 20 September, through Tuesday, 26 September. The TNDLS tested 1259 specimens for the SC NBSL during this period.

Our contingency planning efforts, which began prior to 2016, were beneficial to maintaining the COOP during our three incidents. A natural consequence of a state being impacted by an incident equates to focused efforts on contingency planning, as in our case. We continue to meet regularly to plan and execute strategies to improve the COOP. In this manuscript, we share our experiences in maintaining a COOP and conducting an after-action review (AAR), which is a significant component for identifying strengths, weaknesses, needed improvements, and future actions after incidents. NBS programs engaged in contingency planning may benefit from our accounts, applying some of the lessons learned to their own efforts.

## 2. Materials and Methods

Each NBS program (laboratory and follow-up staff) held internal AARs to include other groups that actively participated in the COOP within the state. These internal groups varied for each incident and included, but were not limited to, laboratory administration, informatics/IT (hereafter IT), epidemiology, procurement, safety, LIMS, and quality. An external AAR was conducted among the impacted laboratory (the laboratory experiencing the incident), the backup laboratory (the laboratory assuming the specimen surge), and each respective state follow-up program. For each AAR, we systematically discussed and documented what went well (strengths), what went wrong (weaknesses), needed improvements, and future actions to better manage any incident (see Appendix A). 

## 3. Results

### 3.1. Collaboration

Strengths: TN, FL, and SC are three states located in the US southeast with some similar characteristics (see Table 1). Stemming from the work carried out by the Southeast Regional NBS & Genetics Collaborative for emergency preparedness, we collaborated prior to 2016 to determine our NBS laboratory capability and planning for the COOP. SC and FL are coastal states and could be impacted by severe weather, making TNDLS ideal as a backup laboratory, since TN is landlocked. FL BPHL is a high-volume laboratory and could easily absorb specimens from TN and SC during an incident. SC is roughly six hours from TN and less than five hours from FL; thus, transit of specimens to the SC PHL by automobile can be accomplished in a relatively short time, with the possibility of sharing employees if the incident demanded. 

Weakness: Although we provided each other support, an incident could impact the entire region, requiring support from an outside partner. Alternatively, because of staffing issues, our current partners may not be able to provide support when it is needed.

Needed improvement: We need to establish additional partnerships inside and outside our region. 

Future actions: Other states in the southeast region, such as North Carolina and Georgia, have expressed interest in a reciprocal relationship and have been invited to collaborate with us. Meetings for NBS contingency planning are being routinely held. Illinois, which is an NBS program with a large birth rate outside of our region, has expressed interest in establishing a reciprocal partnership with us. 

### 3.2. COOP Plan

Strength: We worked occasionally on a shared COOP Resource Guide (hereafter referred to as the plan); therefore, we had some familiarity with each other’s capabilities prior to our incidents. The plan included limited information about the NBS panels, daily volumes, cutoffs, hours of operation, and contact information for key response personnel. 

Weaknesses: While TNDLS had a plan, it was difficult to locate key lab contacts because the plan was stored on a computer at the laboratory as opposed to alternate locations during their incident. The plan was not comprehensive and should have been updated regularly as laboratory and follow-up personnel changed. When the TN follow-up team needed to consult with the FL team, there were no secondary contacts listed for them, and the director for the FL follow-up program could not be reached, though contact was eventually made. The plan did not include coverage for IT or their contacts, so there was difficulty reaching staff over the holiday period.

Needed improvements: We each need a detailed standard operating procedure (SOP) to supplement the plan. The SOP should include everything from specimen accessioning through to reporting of results. Additionally, we need to address what our NBS programs would propose if the program was completely disabled and specimens need to be routed from the point of collection to the backup laboratory. This needs to be included in the plan.

Future actions: We have developed a more comprehensive plan to include the following: laboratory and follow-up primary and secondary contacts, pre-analytical, analytical, and post-analytical testing requirements, reference ranges, testing panels, NBS fees, courier service, results reporting, LIMS, return of specimens, personnel licensure, hours of operations, and more. We need to discuss the COOP with various agencies within our respective states in preparation for an incident. Each of our programs should have dedicated IT staff familiar with the LIMS who can liaise between the laboratory and the vendor for quick fixes. Allocating IT resources during an incident requires advanced preparation; hence, IT staff and other agency staff should be included in contingency planning efforts with partners. To expand the plan for rerouting specimens directly to the backup laboratory, our work with hospitals, providers, and midwives within our respective states should include contingency planning. Internal collaboration at each laboratory is needed to develop an SOP. 

### 3.3. Memorandum of Agreement (MOA)

Strengths: We had signed MOAs between our states that were written explicitly for the COOP in case of an event that disabled our laboratory operations. These MOAs are specific for NBS activities and guarantee coverage in the case of a naturally occurring or man-made disaster or an IT failure. 

Weakness: Although an MOA was in place between SC and TN, due to staff turnover, no one could locate it. 

Needed improvements: The MOA between us should be readily available, as it will be needed during an incident for reimbursement, etc. As we establish additional partnerships outside of our region, an MOA will also be required with these partners. 

Future actions: We plan to begin our MOA revisions at least one year before its expiration so there is no lapse in coverage. We have also ensured that the MOA is widely distributed so key personnel have copies, and a copy is stored in each of our respective office of contracts. 

### 3.4. Simulated Exercises

Strength: We held a simulated exercise to test the COOP in April 2020. We exchanged dummy specimens for testing, and we returned the results to each other in an Excel spreadsheet. 

Weaknesses: Our exercise was meant to evaluate how reporting could happen during an incident. Had we evaluated the entire NBS process, the issues experienced during our incidents would have been discovered. Additionally, we had one exercise even though we had been contingency planning prior to 2016. We also did not fully appreciate that many internal and external partners would be needed for the COOP. This included the vendor of the LIMS system, which had to assist with adding the TNDLS reports into the SC e-reporting system, the commercial couriers that were needed to transport NBS specimens between states, and state procurement, which was needed to make emergency purchases of IT equipment, plane tickets, etc. 

Needed improvement: A full integration of the DBS specimens into the laboratory system from beginning to end during the simulated COOP exercises is needed. All areas involved in sample management, such as sample receiving, accessioning, punching, testing, reporting, and follow-up, should be included in the exercise to test capabilities, identify weaknesses, and expose gaps. 

Future actions: We plan to hold an exercise annually. It will mimic an actual incident as closely as possible to identify areas that warrant improvement. Other departments will be included in these exercises, such as procurement and IT, as their services will be needed during an incident. Discussions are underway for a tabletop exercise between our states with hopes that the exercise will be more robust and streamlined.

### 3.5. Staffing

Strengths: Staff recognized the critical nature of NBS, and they assumed the added responsibility of packaging and shipping specimens, transporting specimens, and testing the specimen surge. Follow-up staff also worked diligently to communicate results to the impacted laboratory in a timely manner. For example, during the first SC incident, the TN follow-up director and the TN case manager coordinator helped their team to call presumed positives. During the second SC incident, the TN case manager coordinator solely handled the responsibility of calling the presumed positives to the SC follow-up team. Those who worked during these incidents are commended for their dedication, as the lives of those identified with a disorder could have been more severely affected without the expeditious testing and follow-up. 

Weaknesses: Lab management rearranged staff schedules during each incident to cover the additional workload, resulting in longer hours or extra workdays. Often, staff earn compensatory time (an hour of leave for every additional hour worked) instead of overtime pay, so little incentive exists to assume any extra workload. Because of the holidays, some staff took scheduled leave; as a result, the area was short-staffed for coverage of the surge in specimens. 

Needed improvements: For future incidents, we should be mindful of the needs of the staff, who may have previously scheduled holiday plans or are working outside of their regular roles. Alternate means of staff compensation are needed to manage future events. 

Future actions: It may be possible to cross-train qualified staff outside of NBS to assist with the accessioning and screening process, which would be helpful in managing incidents that are long term. Cross-training would require cooperation and coordination with the outside sections for success. Discussions with human resources can uncover ways employees can be compensated outside of their regular work schedule. 

### 3.6. Specimen Delivery

Strength: We were able to transport and deliver the DBS specimens to the backup laboratory for testing. 

Weaknesses: Alternate modes of transport are essential, as courier operations may be halted, as was the case during the TN incident. We were not familiar with minor operational details when we chose a courier or commercial shipping company for specimen transport. Several overnight delivery options exist, but we were not aware that some options do not guarantee delivery the next day, especially for items shipped on a Sunday. For example, TNDLS shipped specimens overnight to the FL BPHL, but the specimens shipped on Sunday and Monday arrived on Tuesday morning. To avoid this issue, SC, during their first incident, which spanned the Memorial Day holiday weekend, had staff drive their DBS specimens to TNDLS. Staff also took a commercial flight for faster delivery of the specimens, meaning procurement involvement and approvals became necessary with short notice. For the second SC incident, the NBSL hired a private courier to transport the specimens to TNDLS.

Needed improvement: Screening is already delayed, so the fastest mode to transport specimens to the backup laboratory should be chosen.

Action steps: The contact information for an alternate courier source that can be hired temporarily to deliver specimens to the backup laboratory should be obtained. Different procurement options should be explored if a courier is needed at short notice, such as blanket purchase orders, and if possible, a backup courier should be on state contract. Procurement staff at all levels of the organization should be made aware that emergency approvals are needed during an incident.

### 3.7. Dependency on Technology

Strength: When analytical instruments are interfaced with the DBS punchers and the LIMS, it saves time for worklist creation and reporting results. Although instrument interfaces require a network connection, the transmission of data is often seamless and significantly minimizes laboratory errors. 

Weaknesses: With the lack of an internet connection during the TN and SC incidents, the LIMS could not be accessed, which halted accessioning of the DBS specimens and keying of the demographic information. Punching of the DBS specimens and worklist creations were also affected due to the connectivity issues; hence, no specimens could be tested. When SC PHL lost its network connectivity during its first incident, the entire NBS process was disrupted, and the system shut down.

Needed improvement: Both incidents indicate that a process needs to be developed that allows for continued testing in-house when the internet is down or network connectivity is completely lost. 

Future actions: We plan to work with IT staff and our LIMS vendors to develop and implement an offline procedure that allows punching and testing with the subsequent transfer of data to the LIMS after restoring internet connectivity. The procedure, once written, will be made available to all staff so they know what to do when the need arises. Once in place, work should be carried out with laboratory and IT staff to test these procedures annually for functionality as part of contingency planning. 

### 3.8. Data Entry 

Strength: Data entry staff at the backup laboratories completed demographic entry to facilitate the reporting of results to the impacted laboratory or follow-up program as quickly as possible. 

Weaknesses: The TN hospital and provider information were not in the FL BHPL LIMS, resulting in multiple manual additions. The FL BPHL LIMS would not accept the TN specimen control number (SCN). FL staff could not key the demographic information from the TN forms, so LIMS vendor involvement was required to fix the issue quickly. Many of the NBS forms were handwritten, and only photocopies were sent to the FL BPHL. Due to the poor quality of the copies along with deciphering bad handwriting, data entry was slowed significantly. Furthermore, the lack of clarity contributed to multiple keying errors, which required corrections and mailer reprints several months post-incident. Data entry staff are also accustomed to their own state’s form, so it took time to acquire speed—a challenge for both backup laboratories—during the incidents.

Needed improvements: It is imperative to share our provider databases, as it will be necessary to have this information available if follow-up must be performed by the backup NBS program. We recognize the need to standardize the NBS collection device among our states to expedite the data entry process at the backup laboratory. Only originals will be sent to the backup lab for future incidents.

Future actions: We have started the process of standardizing the NBS collection device with a reformat of the SCN on our respective NBS collection kits to 12 alphanumeric characters (e.g., TN-, FL-, SC-XXXXXXXXXX). This format preserves the SCN as a unique identifier and readily identifies to which state the specimen belongs. Next in the standardization process is to reposition the field placements and standardized naming conventions to help maintain staff cadence and speed when keying. We are also discussing with our respective LIMS vendors the possibility of creating state-specific data entry screens. To facilitate the provider database exchange, the exact layout of the provider information, field lengths, and other necessary information should be shared first, followed by a download with the information arranged in the format preferred by the backup laboratory. Last, the provider databases at each laboratory should be updated and shared at least annually, as the contact information for providers can change, and new providers and birthing facilities are added. 

### 3.9. Specimen Processing

Strengths: The impacted laboratories were able to assign accession numbers to each specimen before shipping specimens to the backup laboratory. For unsatisfactory specimens, the unsatisfactory code was added to the NBS form. In the case of TNDLS, the specimen accession number facilitated matching of the infant to the respective result for follow-up of abnormal results. 

Weakness: TNDLS sent all specimens, including unsatisfactory specimens, to FL BPHL for testing. However, TNDLS soon learned that the FL BPHL does not screen unsatisfactory specimens.

Needed improvements: For future incidents, we need to be aware of each other’s unsatisfactory specimen protocols. We have also decided, for future incidents, to follow the backup laboratory’s protocol for unsatisfactory specimens. 

Future actions: An exchange of our unsatisfactory protocols and the coding used to describe the unsatisfactory types ahead of time will make accessioning easier for the backup laboratory. It would be best to include the backup laboratory’s unsatisfactory code on the demographic portion of the NBS form so the appropriate mailer report can be generated to request a recollection. 

### 3.10. Specimen Volumes

Strength: Both the TNDLS and FL BPHL serving as the backup laboratory could handle the surge in specimens. 

Weakness: The 2022 provisional live birth rates [12] for each of our states (see Table 1) emphasize the sizeable difference in volume among our programs and raise the question of whether TNDLS or SC NBSL could individually handle the FL BPHL volume even for a few days.

Needed improvement: We each need to realistically decide what can and cannot be carried out when serving as the backup lab before an incident occurs.

Future actions: We should each periodically assess the maximum number of DBS specimens that can be reasonably processed daily in addition to our own specimen workload. In making this assessment, we should consider staffing levels, instrument testing capacity, consumables, and supplies. In addition, in this assessment, it should be determined for how long we, as the backup laboratory, can handle the surge in specimens.

### 3.11. NBS Panels

Strength: In the three incidents, we deferred to the backup laboratory’s disorder panel which, in the case of the first SC incident, resulted in an infant being identified with Pompe disease. This was the first case of Pompe for a SC newborn.

Weaknesses: In the case of TN, testing for Fabry and Gaucher could not be performed at FL BPHL, as these disorders are not part of the FL panel. Consequently, TN decided to test for these disorders once the residual specimens were returned. Several providers made inquiries regarding when results would be available. However, it took several days to receive the specimens back in TN because second-tier testing needed to be completed at FL BPHL. In the end, it took approximately three weeks post-incident to complete reporting for these disorders. For future incidents, we will promptly communicate to all providers regarding the COOP activation, outlining panel changes and expectations for additional disorder reporting.

Needed improvement: For future incidents, the prompt return of residual specimens is needed to minimize delays in reporting results for additional disorders not on the backup laboratory’s panel.

Future actions: We will share updates to our panels and changes in methods annually as part of our plan. To minimize delays in screening for additional disorders, TNDLS is exploring the potential of retaining a portion of the DBS specimens in-house to facilitate a shorter turnaround time once the laboratory is operational. We need to explore ways to carry out real-time communication to our hospitals and providers.

### 3.12. Retests and Reference Ranges

Strengths: Infants who had a presumed positive result and who required a specimen recollection were all found; a specimen was recollected and rescreened with results reported. All follow-up actions were closed accordingly by the respective follow-up teams. Providers had no issues with cutoffs even though these differed between labs. Additionally, the laboratory and follow-up teams had cutoffs for all tests performed and testing algorithms and were prepared to explain any differences to providers as needed.

Weaknesses: Some SC specimens that TNDLS reported with a borderline result required that the infant have a recollection and subsequent retest, raising the question as to where the retest should occur, as this had not been initially discussed. Since TN screens for more disorders than SC, a few recollected specimens were sent to TN for repeat testing even after the COOP was deactivated and the SC NBSL was operational.

Needed improvements: Since reference ranges differ between laboratories, the entire rescreening process should be completed at the backup laboratory to maintain consistency with the reporting and testing algorithms. For specimens testing positive for disorders for which the impacted laboratory does not screen, it is obvious that the retest must be carried out at the backup laboratory, and testing algorithms, including higher-tier testing, should be followed.

Future action: We will share our cutoffs and laboratory algorithm changes with each other annually through an update of the plan.

### 3.13. Mailer Format

Strength: Every specimen had a corresponding mailer report that was distributed to the appropriate provider.

Weaknesses: The TNDLS and FL BPHL mailer reports contain the hearing screen and CCHD results, whereas SC NBSL mailer reports do not. Another difference with our mailers is the reporting of individual analyte values versus reporting by disorder panel. The TNDLS and the FL BPHL report results on the mailer by a panel of disorders (e.g., aminoacidopathies). When an analyte (e.g., phenylalanine) is presumed positive, only that analyte value is printed on the mailer under the corresponding disorder panel. On the contrary, if one analyte tested by MS/MS is elevated, the SC NBSL prints the result on the mailer in addition to the remaining MS/MS analytes and their corresponding values, even if these are within normal limits. It also was difficult to differentiate a presumed positive result from one that was within normal limits on some of the mailers.

Needed improvements: Differences in mailer formats could be rectified through standardization. Presumed positive results should be prominently displayed on the mailer, and actions to perform should also stand out. This would make it easier for providers to readily see the results and the corresponding actions needed in response to each type of result.

Future actions: Discussions are needed among us and with our LIMS vendors to investigate the feasibility of standardizing our mailer reports so that, no matter which backup laboratory generates the results, the mailer format is the same or very similar. In the interim, we can reformat our mailers to display presumed positive results in a format that is distinguishable from a normal result.

### 3.14. Access to Results

Strength: The TNDLS granted SC NBSL access to the TNDLS secure remote viewer (SRV), which enabled SC to download their results and distribute them to their providers and birthing centers.

Weaknesses: While SC NBSL had remote access to results, they could not download more than 50 mailers simultaneously. Their results needed to be added to the SC LIMS for long-term access and to meet regulatory requirements, so LIMS vendor involvement was necessary. SC NBSL staff had limited user log-in capabilities to the TNDLS SRV. They had problems searching for specimens because the SC accession number was not captured in the TN LIMS. For the TN incident, the TNDLS did not have remote access to the FL BPHL FNSR system to view or download results. FL BPHL provided paper mailers and sent these by FedEx. Once the paper mailers were received, it took time for TNDLS staff to photocopy the mailers to a shared network drive for long-term storage. To access the stored mailers, TN IT staff wrote a program that renamed each copier file to the appropriate TN specimen accession number.

Needed improvements: SC NBSL needs the ability to download all mailers simultaneously and the ability to have multiple user log-ins. The ability to attach the results from the backup laboratory to each individual specimen in the impacted laboratory’s LIMS is needed to facilitate long-term storage of results and to meet regulatory requirements. To minimize difficulty with specimen look-up and retrieval, each of our LIMS needs a field added to capture the specimen accession number of the impacted laboratory.

Future actions: SC NBSL is currently investigating options for increasing bandwidth to permit larger downloads for printing results. We are each exploring the addition of a field in our respective LIMS to capture the impacted state’s specimen accession number. This will facilitate an easier link of the mailer results to the actual specimen accession number and an easier retrieval of results through a specimen accession number search for those with remote access. TNDLS and FL BPHL are exploring remote access for the download of results as part of our continual contingency planning.

### 3.15. Communications

Strengths: Internal communication between laboratory and follow-up staff during each incident went well. Even before the TN laboratory director provided notification to activate the COOP, the TNDLS and TN follow-up team had already started communicating regarding the possibility of sending specimens to the backup laboratory. Once contact was made with other departments, such as IT and procurement, regarding changing needs, communication was effective, and these parties were available to assist. External communication between the backup laboratory and the impacted laboratory also went well for communicating specimen transport, answering questions regarding specimens, and obtaining results. Communication improved with each incident as staff became more familiar with each other’s processes and operations.

Weaknesses: Differences in follow-up algorithms had not been thoroughly discussed nor were the expectations for when results would be called. For example, the TN follow-up team calls all presumed positive results and borderline results for congenital hypothyroidism to providers. On the contrary, the FL BPHL only sends borderline results by mailer to the providers and does not communicate these results to their follow-up team. Other than presumed positive immunoreactive trypsinogen (IRT) results with one or two mutations, SC follow-up only wanted to know of IRT results > 401 ng/mL with zero variants. TN actively follows up infants with an IRT ≥ 60 ng/mL with no variants and requests a recollection between day of life 10 and 16. TNDLS reports presumed positives to the TN follow-up team seven days a week. Conversely, FL BPHL only reports presumed positives for time-critical disorders on Saturdays to their follow-up team and holds the remaining presumed positives until the next scheduled workday. During the second SC incident, TN follow-up reported presumptive positives to SC follow-up but did not include the SC NBSL, so the SC NBSL did not know to look for mailers for specimens that had to be repeated. Further, because of time differences, some results were called the following day. During the TN incident, TN follow-up received time-critical presumed positive results from the FL team on Thursday (31 December 2020) but, due to the New Year’s holiday closure, did not receive the time-sensitive presumed positive results—though these were finalized—until Monday (4 January 2021).

Needed improvements: An update of follow-up algorithms to current protocols and actions is needed. When calling presumed positive results, we should be cognizant of time zones, especially since the follow-up program and provider offices may be closed due to the time difference. We need to treat presumed positive results as time-critical, since testing has already been delayed.

Future actions: Each of our follow-up teams has agreed to provide updates to their algorithms to each other at least annually. Because specimen analysis is already delayed, no presumed positive result, regardless of time-critical or time-sensitive status, will be held but will be reported to the impacted follow-up program in real time, as this prevents further delays in contacting providers for follow-up actions. During the AAR for the second SC incident, it was communicated to TN that the SC lab needs to know of any presumptive positive results so they can be on the lookout for the corresponding mailers.

## 4. Discussion

NBS programs can never completely prepare for an incident, as there will be unforeseen complications that require flexibility and improvisational approaches to problem solving [13,14]. Nevertheless, when a plan exists, the chances are better for reducing decisions on the fly [14], minimizing downtime, and, ultimately, protecting the most vulnerable. When contingency planning, consideration should be granted to the capabilities that exist to manage an incident, as well as to any existing limitations or barriers that could be encountered [14]. The plan should cover all aspects of specimen testing, protection of records, and follow-up [8]. Specific details regarding follow-up algorithms for calling presumed positives to the handling of corrections should be outlined, as processes vary among NBS programs. The plan should also address supply chain issues, redundancy for supplies [4,13,14], and include a contingency for the collection and transport of specimens [2,4,13].

Factors such as these emphasize that establishing a plan is a collaborative process [13,14] involving multiple parties and agencies. It may be necessary to activate the COOP agency-wide to assist efforts and provide resources in real time. NBS needs should be assigned elevated priority to ensure continuity [6]. The plan should also be tailored to meet the needs of varying situations [14], e.g., complete shutdown of the laboratory and follow-up versus maintaining some level of functionality. In developing a plan, the geographic location of the programs involved should also be considered [2] because different approaches may be needed to manage the COOP depending on the type of incident. Once the plan is developed, it is a living document, meaning that as capabilities and other aspects of the NBS program change, so should the plan. Therefore, the plan should be updated to capture changes with a recommended frequency of at least every six months to a year [6,14].

Another aspect of contingency planning that is crucial is an MOA, as it helps to ensure that laboratory testing will continue, albeit at a backup laboratory [2,6,10,13]. Without an MOA, invoicing or reimbursement for services can be challenging; therefore, it is beneficial to establish an MOA prior to an incident to simplify these processes [1,6]. An MOA should clearly outline the scope of services, funding or reimbursement for services, transportation and delivery of specimens, chain of custody needs, contact persons, liability, and terms and termination of the agreement and should be executed with appropriate signatures [6,10,13]. Our experience in developing MOAs revealed that the process for approvals can be lengthy. MOAs need to be reviewed by each state’s legal department for appropriate content and language. The document will have several revisions before it is ready to be signed.

Good preparation to manage an incident means assessing the plan [10] every six months [14], or at least annually [6,7], with subsequent debriefing and documentation of the assessment [8,14]. An exercise should fully test the plan to identify weaknesses and expose gaps [7,10,14]. Through exercises, all involved will know their roles and responsibilities, and these allow for built-in redundancy, as some staff may need to assume overlapping responsibilities during the entire COOP [10]. With the added COOP responsibilities, staff may experience stress from changes to their normal routine. Laboratory leadership should be sensitive to staff needs, especially if they too are impacted by the incident [2]. Staff availability should also be considered, as their willingness to respond is critical to an effective response [9]; however, any long-term changes in routines and a constant increase in workload can negatively impact staff and lead to employee burnout [5].

Laboratories rely heavily on automation and technology; without them, it is almost impossible for laboratories to operate. Contingency planning should consider offline operations or even the use of a wireless mesh network (WMN) so that testing in-house remains uninterrupted. A WMN functions independently of cellular or Wi-Fi, electrical grids, and traditional communication infrastructures [15]. Having a WMN has proven effective in disaster relief situations where communications and connectivity were crucial for the COOP [15,16]. Regarding LIMS functionality, cloud computing platforms allow data storage with unlimited space, backup, and reliable service; however, security, privacy, and rapid data recovery may pose issues during an incident [17].

Web-based reporting is timely and may be ideal for communicating results since landlines may be disabled during an incident [18]. Reporting variations can be minimized through web-based reporting, which facilitates the providers’ access to the results [18]. A more optimal approach would be electronic reporting through Health Level 7 (HL 7) messaging [19,20]. Electronic data exchange or interoperability between laboratories has proven to be beneficial before, during, and after an incident [2], as it facilitates not only result reporting but also laboratory orders [19]. Interoperability would be especially beneficial to decrease the amount of time spent entering demographic information at the backup laboratory.

An additional opportunity exists to harmonize our reference ranges to achieve uniformity in reporting. Harmonization would allow cutoffs and result values across our laboratories to hold the same meaning [21,22]. Harmonization of reporting practices is also an opportunity to reduce variability ensuring NBS results are readily available to providers [18]. Harmonization of technology and methodologies has been recommended as a way to contribute to consistency and improved coordination for the COOP [2]. It has been noted that few states may be able to absorb a significant increase in volume, even for a short period [6,10]. Outside of state licensing requirements, harmonization could facilitate employee sharing to balance the workload, as employees would already be competent and require little time to become acclimated to the backup laboratory’s routine [6,10].

Ideally, when contingency planning, high-volume laboratories should consider using a backup laboratory that matches their daily volume. It may be possible to split specimens between a few smaller backup laboratories to ensure coverage when there is not a large backup laboratory that can match volumes or cannot assume the surge. A caveat to splitting specimens is managing communications with more than one program; however, splitting specimens could aid in covering all disorders on the impacted laboratory’s panel if the backup laboratories collectively screen for those disorders. It is important to note that the surge in specimens could exhaust the backup laboratory’s consumables and reagents, so vendors should be alerted of the incident to quickly replenish supplies. The surge may impact turnaround times at the backup laboratory, causing these metrics to fall outside of the national recommendations for disorder reporting [20,23].

If possible, laboratories with reciprocal partnerships should preferably match disorder panels to facilitate testing of all disorders [10]. Matched panels minimize confusion for providers and specialists, as opposed to receiving more or less screening results than is customary. When panel matching is impossible, the impacted lab should defer to the backup laboratory’s panel even if the backup laboratory screens for more disorders, since not doing so is disruptive to routines and could cause issues with integrating the surge of specimens into the workload. A drawback to consider is that a disorder may be detected with which the follow-up staff may be unfamiliar. As a result, difficulty in recommending actions or referring the infant to an appropriate specialist may arise [6]. Scenarios such as these emphasize how imperative it is for the impacted laboratory to communicate that COOP has been activated, that results will be delayed, and that there is a difference in the NBS panel, as well as the cutoffs and reporting format [6]. Furthermore, if the impacted laboratory screens for more disorders than the backup laboratory, the impacted laboratory should decide whether to screen for the additional disorders later and clearly communicate the decision to providers [10].

As a nation, the need for more robust public health laboratory infrastructure and redundancy in testing is noted [1]. To meet that need, select NBS laboratories structured much like the Antimicrobial Resistance Laboratory Network, which provides specialized laboratory test services at the regional level, should be considered [1]. Having several NBS laboratories strategically placed across the US means these laboratories can devote more time, staff, and resources to contingency planning and surge capacity. This NBS network would mitigate the need for laboratories to take on the burden of testing for another state laboratory when they are unprepared to do so. Assistance from the Centers for Disease Control and Prevention and the Association of Public Health Laboratories would be needed to establish this type of network for NBS response.

## 5. Conclusions

Our three incidents highlight the importance of contingency planning for NBS COOP. Contingency planning is a continual process and should be thought of as a marathon and not a sprint. It cannot be left to chance and requires conscious time and effort from multiple partners. It should be an integral part of laboratory operations, just like other routine processes. Frequent communication is necessary among NBS programs to develop a plan that becomes a living document that should be updated and exercised periodically. By doing so, any incident, be it short or long term, can be successfully managed.

## Figures and Tables

**Table 1 IJNS-10-00055-t001:** State NBS program comparison.

	FL	SC	TN
Annual births ^1^	224,226	57,775	82,262
Average daily volume ^2^	900	200	300
Core conditions screened (38)	35	36	36
Testing personnel	23	19 ^3^	23 ^3^
Follow-up housed within the laboratory	No	No	Yes
Short-term follow-up personnel	17	4	16
State licensing is required for testing personnel	Yes	No	No
Days of operation	6 days/week	6 days/week	7 days/week

Notes: ^1^ 2022 provisional data [12]; ^2^ includes recollections; ^3^ includes data entry staff.

## Data Availability

Data are available on request to the authors.

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
