# Peer review of "Continuity of Operations in Newborn Screening: Lessons Learned from Three Incidents"

_2409-515X, 2024, doi:10.3390/ijns10030055_

Round 1

Reviewer 1 Report

Comments and Suggestions for Authors

I have attached my comments as a file.   

They are also here but the formatting may be problematic.

Thank you for allowing me to review the article Continuity of Operations in Newborn Screening:  Lessons Learned from Three Incidents.  I think this is a great article that will be beneficial for IJNS readers.  I would appreciate, however, if the authors could restructure the results and discussion.  While I appreciate the “positive” “improvements needed” and “action steps” for many parts of section 3, I am missing the detail of what happened during the incidents to understand these; some of these details don’t appear until the discussion (e.g. lines 353-362).  Here are some suggestions to make it easier for the reader: (1) add a what went wrong section to your existing 3 sections (positives could become a “what went right” section) or (2) merge the discussion and the results.  You have a lot of info about the incidents in your discussion.  Right now your discussion is too long and I think this is because you have so much information about the incidents in there.

This feels like a quality improvement project (or the first step in one). The SQUIRE 2.0 guidelines might help you figure out how to bring the information together so that it is easier for the reader to follow your suggestions (https://www.squire-statement.org/index.cfm?fuseaction=Page.ViewPage&PageID=471).

I think there is a lot of value in the information that is presented but I need more detail of the incident sooner; I should not learn anything new about the incidents in your discussion.

Some more specific feedback:

1.      Line 1. “  . . can strike at any moment and with little warning if any.”  You don’t need the words “if any”.

2.      Lines 20-22.  You make it sound as though NBS is just bloodspots which is not accurate.  You either need to address all of NBS or say you are only focusing on bloodspot screening and why.

3.      Lines 60-62.  This sentence on the ability to maintain the COOP is awk.  More detail might help.  What do you mean when you say “no doubt due to our contingency planning throughout the years”.  I’d rather hear more about that and less about NBS background (e.g. we know it is the most successful public health program in the US, We know how many babies are typically screened in the US).  You can take some of that detail out (lines 13-20) and put in more detail on your contingency planning relationship either in background, methods, or results. The info in section 3.2 is helpful.  I’d like to see that prior to the problems.

4.      Line 82: I believe it should read “state of TN” vs. “State of TN”

5.      In section 3.1 you use “Tennessee” and “TN”.  I think you can just use TN.

6.      Line 89:  I think FL testing 1,109 DBS for TN right?  Or is that total – TN and FL’s DBSs?

7.      While I appreciate the picture of the part that caused the SC outage, I am not sure that level of detail is needed.  Either take it out or help me understand what it is important.

8.      Line 110:  Again, I assume the 1,259 specimens tested were for SC right?  Or is that the total number tested by TN during that time period?

9.      While I appreciate the details on the incidents, the description does not help me understand all of the results – how did you arrive at your positives, improvements needed, and action steps based on these 3 incidents?  They don’t seem connected to your collaboration results.

10.  Line 122:  I don’t follow the conclusion “we recognize the need to include additional partners”.  Why?  What need is not met by your 3 state partnership? What failed in operating your COOPs that shows there is a need to make this partnership bigger?

11.  Lines 133-143: What happened in your 3 incidents to lead you to include this other information in your plan other than the need for IT? 

12.  Lines 144-145:  Do your 3 programs use the same LIMS?  Do the IT staff have to be familiar with the other state’s LIMS or does that not matter?  If it does matter, can you edit Table 1 to say what LIMS and follow-up IT system each program uses?

13.  Line 149:  are you missing the word “the”  “the COOP”?

14.  Lines 147-148.  Are the monthly preparedness calls within the state system or between the state systems?

15.  Lines 161-163.  Can you speak more the “easily retrievable” part of the MOA?  Was it for your instances? Where was it stored?  What can be done or thought about to make sure it is easily retrievable?

16.  Line 176 – I think you need to say “exercises should be held at least annually”.  Take out “greater frequency” since we don’t know what the frequency was.

17.  For 3.6 Staffing you talk about the dedication of staff and being mindful of previous plans. Was this especially true for the incident close to Christmas?

18.  For 3.7 – How far out of ACHDNC recommended timing were you?  Can you provide more detail as to why?  It sounds like it took a while to get a backup courier but you don’t state that.

19.  Sections 3.12 & 3:15 had good detail in it about the incidents.  I’d like to see that for all the results.

20.  Lines 296 to 297: How will you share the cut-offs annually?  Email, update COOP?

21.  I don’t have specific suggestions for the discussion just yet because I am hoping the detail about the incident in the discussion gets moved up.

Reviewer 2 Report

Comments and Suggestions for Authors

The manuscript by Dorley, et al., describes the value and utility of COOP planning in newborn screening. The manuscript could be improved by addressing the following:

1) Introduction. The introduction does not provide sufficient background to describe why cross-state COOP planning is difficult in NBS. More background on the state-based nature of NBS and differences across state lines should be included. Likewise, there is too much description of wording choice (i.e., incident) in the manuscript that seems unnecessarily detailed. Please also check references as some do not seem to be appropriate for the sentence they are attributed to.

2) Throughout. There are several instances where acronyms are provided in parentheses, but never used again. This is unnecessary. There are also long run-on sentences that make it hard to follow some ideas. 

3) Figure 1 is unnecessary

4) Section 3.2. Provide more detail on what other partners are needed and how many other states are on the calls. 

5) Table 1. Unclear what "licensure required" is referring to? Follow-up staff? Clinicians? Other?

6) General Formatting. Consider using terms like "Strengths," "Needed Improvements", "Future Actions". It is a bit confusing the way these sections are written as some are written as though they are geared towards the 3 state authors and some as though they are geared more overarchingly to the NBS community

7) DiscussionThe discussion is quite long and seems largely redundant to the Results section. Consider condensing this to focus on key takeaways of the paper. 

Comments on the Quality of English Language

Language is a bit hard to follow with run-on sentences.

Round 2

Reviewer 1 Report

Comments and Suggestions for Authors

The authors did an excellent job of responding to my concerns. The way this article is restructured makes it much easier to follow and to understand the recommendations. Thank you to the authors for that - it was not an easy task.

I have one small edit.  On Line 237 you have "incident.-".  You need to get rid of the line/dash.

This is an excellent article and will help other states think more thoroughly about their COOPs.